# LEARNING TWO-TIME-SCALE REPRESENTATIONS FOR LARGE SCALE RECOMMENDATIONS

## ABSTRACT

We propose a surprisingly simple but effective two-time-scale (2TS) model for learning user representations for recommendation. In our approach, we will partition users into two sets, active users with many observed interactions and inactive or new users with few observed interactions, and we will use two RNNs to model them separately. Furthermore, we design a two-stage training method for our model, where, in the first stage, we learn transductive embeddings for users and items, and then, in the second stage, we learn the two RNNs leveraging the transductive embeddings trained in the first stage. Through the lens of online learning and stochastic optimization, we provide theoretical analysis that motivates the design of our 2TS model. The 2TS model achieves a nice bias-variance trade-off while being computationally efficient. In large scale datasets, our 2TS model is able to achieve significantly better recommendations than previous state-of-the-art, yet being much more computationally efficient.

## 1   INTRODUCTION

A hypothetical user's interaction with recommendation systems gives us *diminishing* returns in terms of its information value in understanding the user. For an *active* user who has lots of historical interactions, she is typically well understood by the recommender, and each new interaction gives relatively little new information. In contrast, for an *inactive* or *new* user, every additional interaction will provide interesting information for understanding this user. Therefore, the representations for active and inactive users should be updated differently when a new interaction occurs. Figure 1 illustrates such information diminishing phenomenon, where the amount of change in user embedding from $\phi^t$ to $\phi^{t+1}$ due to an additional interaction is decaying. One can select a particular threshold $t^*$ for the number of interactions, above which the users can be categorized to active users, and below which inactive users. Roughly active users' embeddings evolve slowly as a function of the number of interactions, while inactive users' embeddings evolve fast. Hence a two-time-scale embedding evolution.

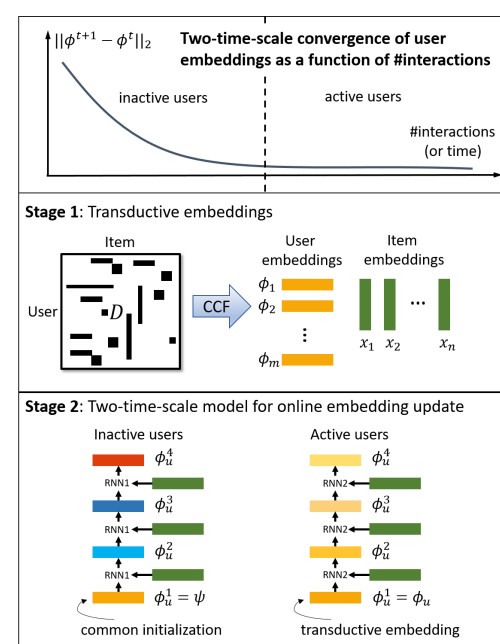

Figure 1: Two-time-scale convergence of user embeddings motivates us to design a two-stage method, where the first stage estimates transductive embedding and the second stage learns two different RNNs for active and inactive users respectively.

Apart from the time-scale difference in temporal dynamics, the simultaneous presence of active and inactive users also presents other modeling and computational challenges. On the one hand, active users lead to long sequences of interactions and high degree nodes in the user-item interaction graph. Existing sequence models, such as RNN models, have some limitations when dealing with *long-range* sequences, due to the difficulty in gradient propagation. Moreover, graph neural network-based models become *computationally inefficient* due to the intensive message passing operations through high-degree nodes introduced by active users. On the other hand, predicting preferences of inactive or

new users (also known as the *cold-start* problem) is a challenging few-shot learning problem, where a decision needs to be made given only a few number of observations. To address various challenges imposed by the presence of two types of users, we leverage the different dynamics of these users and propose **(i) a two-time-scale (2TS) model** and **(ii) a two-stage training algorithm**.

**2TS model.** Based on the number of observed interactions, we partition the users into two sets: active and inactive users. Our 2TS model (Fig. 1) update the embeddings of active users and inactive users by two RNNs with independent parameters, in order to respect the two-time-scale nature. Moreover, the initial embeddings of inactive users are represented by a common embedding $\psi$, which is *shared* across all inactive users. Therefore, the overall model for inactive users is *inductive*, in the sense that the learned model can be applied to unseen users. In contrast, the initial embedding of each active user is a *user-specific* embedding $\phi_u$, which is also called *transductive* embedding. Such embeddings are very expressive, which can better express users with a long history.

**Two-stage training.** In stage 1, we first learn transductive user embeddings $\phi_u$ and transductive item embeddings $x_i$ using a classical collaborative filtering method. Then we fix these embeddings, and in stage 2, we will learn the parameters of the two RNNs and a common initialization $\psi$ for inactive users. It is notable that the transductive embeddings for inactive users are abandoned in stage 2. Only those for active users are finally used in the 2TS model. Besides, for active users, we do not use all interaction data to learn the RNN since their transductive embeddings have already encoded the information of their history. We only use a small number of last clicked items to learn the adaptation for active users, which improves the efficiency of the training process.

The proposed 2TS model and the two-stage training algorithm lead to a few advantages:

- **Bias-variance trade-off.** The differential use of transductive and inductive embeddings for the two RNN models allows 2TS to achieve a good overall bias-variance trade-off. We theoretically analyze such trade-off in Section 2 through the lens of learning-to-learn paradigm for designing online learning (or adaptation) algorithms. Our theory shows that there exists an optimal threshold to split users to achieve the best overall excessive risk.
- **Encode long-range sequence.** The transductive embeddings $\phi_u$ for active users are user-specific vectors, so they can memorize the user's long-range history during the training, without suffering from the difficulty of gradient propagation. The RNN on top of these transductive embeddings is only used for adaptation to recently engaged new items.
- **Computational efficiency.** The efficiency of our method on large-scale problems mainly comes from two designs in the algorithm. First, stage 1 learns the transductive embeddings of active users and items, which contain a large number of parameters. However, it is fast since it does not involve any deep neural components and the loss is simply a convex function. Second, stage 2 only learns the RNNs which contain a small number of parameters, and the RNN for active users is only trained on a few last engaged items, which cuts off the long sequences. Experimentally, our method reveals to be much more efficient than the baselines on large-scale datasets.

We summarize the contributions of this paper as follows:

- To explain the intuition and motivation of the 2TS model, we provide theoretical analysis on a simplified setting, which rigorously argues the need for differential use of transductive and inductive embeddings for active and inactive users (Section 2).
- Motivated by the analysis, we design the 2TS model and a two-stage training method, for practical use (Section 3). The proposed method is applied to two large-scale benchmark datasets and compared comprehensively to various baseline models, spanning a diverse set of categories, which shows that our method is advantageous in terms of both accuracy and efficiency (Section 5).

## 2 THEORETICAL MOTIVATION: WHY TWO-TIME-SCALE MODELS?

We will first present the motivation for designing the 2TS model, through the lens of online learning and stochastic optimization. Our analysis quantitatively reveals that *(i)* the embeddings for active and inactive users evolve in different time scales, and *(ii)* two different online learning algorithms for active and inactive users respectively can lead to a better overall estimation of user embeddings.

Our analysis will be carried out in a learning-to-learn setting, where online learning algorithms need to be designed to tackle a family of tasks for estimating the embedding vector of a user. Though this idealized setting can not cover all aspects of real-world recommendation problems, it leads to

clear insights on the 2TS behavior of active and inactive users and the benefits of using two different online algorithms for these two respective user groups. These insights also motivate our practical implementation of the 2TS model using deep learning in Section 3.

## 2.1 SETTING: LEARNING-TO-LEARN

Our setting consists of three components: the estimation task for an individual user, the distribution of tasks for a family of users, and the online algorithms which we want to design.

**Individual Task.** We associate each user $u$ with a ground truth embedding $\phi_\mu^* \in \mathbb{R}^d$, which can be thought of as a vector representation of a user's preference. This embedding is defined by a distribution $\mu$ over user's clicks over items $(\boldsymbol{x}, y)$, where $\boldsymbol{x} \in \mathcal{X}$ represents the item embedding which we assume is bounded, i.e., $\|\boldsymbol{x}\|_2 \leq B_x$, and $y \in \{0, 1\}$ indicates whether the item is clicked. They follow the user-specific distribution $(\boldsymbol{x}, y) \sim \mu$. More specifically, the ground truth user embedding is defined as the minimizer of the *expected risk* according to a regularized logistic loss

$$\phi_\mu^* := \arg\min_{\phi \in \Phi} \mathcal{R}_\mu(\phi), \quad \text{where } \mathcal{R}_\mu(\phi) := \mathbb{E}_{(\boldsymbol{x}, y) \sim \mu} \ell(\phi, \boldsymbol{x}, y),$$
$$\text{and } \ell(\phi, \boldsymbol{x}, y) := -y\boldsymbol{x}^\top\phi + \log(1 + \exp(\boldsymbol{x}^\top\phi)) + \tfrac{c}{2}\|\phi\|_2^2, \quad (1)$$

where $c > 0$ is some regularization constant and $\Phi := \{\phi \in \mathbb{R}^d : \|\phi\|_2 \leq B_\phi\}$.

Typically, we do not have access to the distribution $\mu$, but a sampled set of $T$ observations $z_{[T]} := \{(\boldsymbol{x}_1, y_1), \cdots, (\boldsymbol{x}_T, y_T)\} \sim \mu^T$, which can be used as training samples to obtain an initial estimate $\phi(z_{[T]})$ of the user embedding. We assume $\phi(z_{[T]})$ is obtained by applying stochastic gradient descent (SGD) to the loss $\ell$ over $z_{[T]}$ in this section, but this training stage is not limited just to the SGD algorithm. It is expected that with more samples the estimation will be closer to the ground truth $\phi_\mu^*$. The estimation $\phi(z_{[T]})$ is modeling the offline training stage of a recommendation system.

**A Distribution of Tasks.** We consider a distribution of users, by assuming that the user-specific distribution $\mu$ is sampled from a meta-distribution $\mu \sim p_u$. Furthermore, the number of observed interactions for a user, denoted by $T \sim p_T^\alpha$, is a random variable that follows a power law distribution with density $p(T) \propto (T + 1)^{-\alpha}$. The power law distribution models the fact that there will be lots of users with very few interactions, and very few users with many interactions.[1] A key assumption of our model is that the variance of the ground truth user embeddings is small. That is

$$\text{Var}_{\boldsymbol{m}} = \mathbb{E}_{\mu \sim p_u}\|\phi_\mu^* - \boldsymbol{m}\|_2^2 \leq r, \quad \text{where } \boldsymbol{m} = \mathbb{E}_{\mu \sim p_u}\phi_\mu^*. \quad (2)$$

The assumption of small variance is critical in the sense that it will allow us to aggregate information from inactive users to obtain better estimations of their user embeddings.

**Online Algorithm Design Problem.** Our goal is to design online adaptation algorithms for this distribution of users, such that the overall excessive risk of the online adaptation across all users is small. This is to model the online sequential recommendation stage where new user-item click information is incorporated after system deployment. Note that we are not restricted to design a single online learning algorithm for all users. In fact, we will show later that 2 online learning algorithms can actually lead to better risk bound than 1 online learning algorithm.

In this algorithm design problem, each user $\mu$ corresponds to an online learning task, where items arrive sequentially. Starting from an initial embedding $\phi_\mu^1$, an online algorithm updates the user embedding whenever it observes a new user-item interaction $(\boldsymbol{x}_t, y_t) \sim \mu$:

$$\phi_\mu^{t+1} \leftarrow \text{Update}(\phi_\mu^t, \boldsymbol{x}_t, y_t), \quad (3)$$

and then it applies $\phi_\mu^{t+1}$ to the next item $\boldsymbol{x}_{t+1}$, and suffers the loss $\ell(\phi_\mu^{t+1}, \boldsymbol{x}_{t+1}, y_{t+1})$ in Eq. 1. The **excessive risk** of this online algorithm after encountering $N$ sequential samples is

$$\tfrac{1}{N}\sum_{t=1}^N \ell(\phi_\mu^t, \boldsymbol{x}_t, y_t) - \ell(\phi_\mu^*, \boldsymbol{x}_t, y_t).$$

The design problem involves (1) the initialization for the algorithm and (2) the update step of the algorithm. One obvious choice is to use the user-specific embedding $\phi(z_{[T]})$ estimated in the offline training phase as the initial embedding $\phi_\mu^1$ for the online algorithm. However, is this estimation the best choice for the initial embedding $\phi_\mu^1 = \phi(z_{[T]})$? Is there a better choice? How should the initialization depend on $T$? We answer these questions in the next subsection.

---

[1]This assumption is sensible since Fig. 5 shows that $T$ approximately follow a power law in real-world datasets.

## 2.2 ANALYSIS AND ITS IMPLICATIONS

We focus on designing online gradient descent (OGD) algorithms for Eq. 3, where the update step is $\phi_\mu^{t+1} \leftarrow \text{Proj}_\Phi \left[ \phi_\mu^t - \gamma \cdot \partial \ell(\phi_\mu^t, \boldsymbol{x}_t, y_t) \right]$. In our theoretical analysis, we will focus on designing only the initial embedding for the algorithms and the learning rate $\gamma$ to gain insights, and deferred the design of more components of the algorithms to later practical neural implementation.

A key aspect of our analysis is that we will consider a threshold $t^* \in \mathbb{N}$ that divides the users into two groups: **active users** with offline training sample $T \geq t^*$ before online adaptation and **inactive users** with $T < t^*$. Under such split of the users into two groups, we will design 2 online learning algorithms, one for the active user group and the other for the inactive user group. Furthermore

- **Different initialization.** The initial embeddings $\phi_\mu^1$ for the two online learning algorithms are designed as: for active users, their learned embeddings $\phi(z_{[T]})$ are used as the initialization, and for inactive users, they use a common embedding as the initialization. Mathematically,

$$\textbf{Active User: } \phi_\mu^1 = \phi(z_{[T]}) \text{ if } T \geq t^* \text{ and } \textbf{Inactive User: } \phi_\mu^1 = \boldsymbol{m} \text{ if } T < t^*, \quad (4)$$

  where $\boldsymbol{m}$ represents the common initialization. We will later show in Theorem 2.1 that the optimal choice of the common initialization is the mean $\boldsymbol{m} = \mathbb{E}_\mu \phi_\mu^*$, and we can also analytically find the optimal threshold $t^*$. The intuition is that, for inactive user, if we start from the estimation $\phi(z_{[T]})$ and make online adaptation for the new items, it will not perform well due the large discrepancy between $\phi(z_{[T]})$ and the ground truth $\phi_\mu^*$. However, since $\phi_\mu^*$ is distributed around the mean $\boldsymbol{m}$, we can aggregate the offline observations from inactive users to estimate $\boldsymbol{m}$. Then it may be better off to use the estimated $\boldsymbol{m}$ as the shared initial user embedding for the inactive users. In contrast, for active users, $\phi(z_{[T]})$ can be a better initialization than $\boldsymbol{m}$ since it is already very close to $\phi_\mu^*$.
- **Different learning rate.** The learning rates $\gamma$ are designed to be different for the two online learning algorithms. As later showed in Theorem 2.1, the optimal learning rate for active users is smaller than that for inactive users, and leads to a better overall excessive risk. The intuition is that, for an inactive user, each additional observation provides us a lot of information about this user, and we want to employ a larger learning rate to fully ingest such information. In contrast, for an active user, the additional observation provide diminishing amount of information about this user, and we can use a smaller learning rate.

In the theorem below, we show precisely that different initialization and learning rate for the active and inactive users can lead to a smaller expected excessive risk. Furthermore, there is a theoretical optimal choice of the threshold $t^*$ to achieve this. A more detailed statement of the theorem and the proof of the theorem are provided in Appendix A.

**Theorem 2.1.** *Assume the setting and notation in Section 2.1. Assume $\phi_u^t$ is updated by OGD with step size $\gamma$. With the initialization strategy in Eq. 4, the expected excessive risk is bounded by*

$$\mathbb{E} \frac{1}{N} \sum_{t=1}^N \ell(\phi_\mu^t, \boldsymbol{x}_t, y_t) - \ell(\phi_\mu^*, \boldsymbol{x}_t, y_t) \leq (2\gamma N)^{-1} \left( \beta(t^*) + \gamma^2 B^2 N \right), \quad (5)$$

*where $\beta(t^*) = \frac{\zeta(\alpha+1, t^*+1)}{\zeta(\alpha,1)} \bar{Q} + \left(1 - \frac{\zeta(\alpha, t^*+1)}{\zeta(\alpha,1)}\right) \text{Var}_{\boldsymbol{m}}$, $B = B_x + cB_\phi$, $\zeta$ is the Hurwitz zeta function, and $\bar{Q}$ is a constant larger than $\text{Var}_{\boldsymbol{m}}$. Optimizing the step size $\gamma = B^{-1}\sqrt{\beta(t^*)/N}$, we have*

$$\mathbb{E} \frac{1}{N} \sum_{t=1}^N \ell(\phi_\mu^t, \boldsymbol{x}_t, y_t) - \ell(\phi_\mu^*, \boldsymbol{x}_t, y_t) \leq B\sqrt{\beta(t^*)/N}. \quad (6)$$

*Moreover, the optimal choice of the threshold $t^*$ is*

$$t^{**} = \arg\min_{t^* \geq 0} \beta(t^*) = \lfloor \bar{Q} / \text{Var}_{\boldsymbol{m}} \rfloor - 1. \quad (7)$$

**Implication 1: A differential design for the initial embeddings is better.** If all users are initialized by their user-specific embeddings $\phi(z_{[T]})$, then it corresponds to the case when $t^* = 0$. However, it is a less optimal choice than $t^* = t^{**}$, and the gap between them can be quantitatively characterized by $\beta(0) - \beta(t^{**}) = \sum_{t=1}^{t^{**}} (\frac{\bar{Q}}{t} - \text{Var}_{\boldsymbol{m}}) t^{-\alpha}$. Since the excessive risk is proportional to $\sqrt{\beta(t^*)}$, the gap shows the advantage of the differential initialization for active and inactive users in Eq. 4. Furthermore, if $\text{Var}_{\boldsymbol{m}}$ is smaller, the gap and the advantage of differential initialization are larger.

**Implication 2: The optimal step sizes for active and inactive users are different.** In the above theorem, the optimal step size $\gamma$ for the online algorithm is optimized over all users. However, if we

consider using two different step sizes for the group of active users and the group of inactive users, the optimal step sizes for these two groups will be (see derivations in Appendix A.2)

$$\gamma_{act} = \sqrt{\frac{\zeta(\alpha+1,t^*+1)\overline{Q}}{\zeta(\alpha,t^*+1)N}} \text{ for active users, and } \gamma_{in} = \frac{1}{B}\sqrt{\frac{\text{Var}_{\boldsymbol{m}}}{N}} \text{ for inactive users.}$$

It is easy to verify that $\gamma_{act} < \gamma_{in}$, which suggests that embeddings of active users should evolve with a smaller step size than inactive users. This is consistent with our intuition for the 2TS model.

## 3 METHODOLOGY FOR PRACTICE

In this section, we will describe our concrete methodology, including *(i)* the parameterization of the two-time-scale (2TS) model and *(ii)* a two-stage training algorithm for learning this model efficiently and effectively. The design is motivated and guided by the analysis in Section 2, though more advanced techniques are incorporated in order to achieve the state-of-the-art performance.

Instead of following the OGD algorithm in Section 2, we will use more flexible models for the online update dynamics of user embeddings, as in the meta-learning framework (Ravi & Larochelle, 2016). In particular, we will use two different RNNs, which we call two-time-scale (2TS) model, to model the dynamics of active and inactive users respectively, which correspond to the update steps in Eq. 3.

To learn the 2TS model, we first perform a generalized collaborative filtering method to obtain *transductive* user and item embeddings, which is similar to the training phase in Section 2, except that item embeddings also need to be learned from data and a more advanced loss function is adopted. The learned transductive embeddings will be fixed in stage 2. Transductive embeddings for active users will be used as the initial embeddings during the training of RNNs, while for inactive users a common initialization will be learned together with the RNNs.

### 3.1 NEURAL TWO-TIME-SCALE (2TS) MODEL

Consider a set of users $u \in \mathcal{U} := \{1, 2, \cdots, U\}$ and items $i \in \mathcal{I} := \{1, \cdots, I\}$ in a system. In our model, users are divided into two sets by the *number of observed interactions $T$* in the training set. With a threshold $t^* \in \mathbb{N}$, we consider users with more than $t^*$ observed interactions as active users $\mathcal{U}_{act}$, and those with less than $t^*$ interactions as inactive users $\mathcal{U}_{in}$, which include new users.

Our two-time-scale model consists of two RNNs, one for active users and the other for inactive users. They can be thought of as two different online adaptation operators for active and inactive users respectively. Furthermore, these two RNNs will also differ in the way in whether they have user-specific parameters and how they are initialized. Assume for the moment, we have already learned an initial set of transductive user embeddings $\{\boldsymbol{\phi}_u \in \mathbb{R}^d : u \in \mathcal{U}\}$ and item embeddings $\{\boldsymbol{x}_i \in \mathbb{R}^d : i \in \mathcal{I}\}$, which will be explained in later two-stage training methods.

The RNN for inactive or new users is purely inductive in the sense that it does not contain user-specific parameters. We will **not** use user-specific transductive embeddings $\boldsymbol{\phi}_u$ in this RNN. Instead, the RNN will start with a learned **common initialization** $\boldsymbol{\psi} \in \mathbb{R}^d$, and then be updated by RNN-based adaptation operators every time a new interaction is observed. More precisely, for an inactive user who clicks on items $(i_1, \cdots, i_T)$ sequentially, the embedding will be updated sequentially as:

$$\textbf{inactive users } u \in \mathcal{U}_{in}\textbf{:} \quad \boldsymbol{\phi}_u^0 = \boldsymbol{\psi}; \quad \boldsymbol{\phi}_u^{t+1} = \texttt{RNNcell}_{\Theta_1}(\boldsymbol{\phi}_u^t, \boldsymbol{x}_{i_t}). \tag{8}$$

For active users, we will view their transductive embeddings as the memories for their long-term history. Different from inactive users, the initial embeddings in the RNN for active users will be set to be the user-specific transductive embeddings $\boldsymbol{\phi}_u$. Then the RNN-based adaptation operator will update this initial transductive user embeddings if there are more interacted items:

$$\textbf{active users } u \in \mathcal{U}_{act}\textbf{:} \quad \boldsymbol{\phi}_u^0 = \boldsymbol{\phi}_u; \quad \boldsymbol{\phi}_u^{t+1} = \texttt{RNNcell}_{\Theta_2}(\boldsymbol{\phi}_u^t, \boldsymbol{x}_{i_t}). \tag{9}$$

The inactive and active user models in Eq. 8 and Eq. 9 constitute our two-time-scale model (2TS model). The model consists of several sets of parameters, including the transductive embeddings for active users $\{\boldsymbol{\phi}_u \in \mathbb{R}^d : u \in \mathcal{U}_{act}\}$ and items $\{\boldsymbol{x}_i \in \mathbb{R}^d : i \in \mathcal{I}\}$, the two sets of RNN parameters $\Theta_1$ and $\Theta_2$ respectively, and the additional common initialization $\boldsymbol{\psi}$ for the inactive user RNN. These parameters will be learned differently using a two-stage method described in the next section.

## 3.2 Two-stage Training

Let $\mathcal{D}$ be the training dataset that records the interactions between users and items as an ordered sequence where the $t$-th interaction between some user $u$ and item $i$ is denoted as $e_t := (u_t, i_t)$. Our training method for the two-time-scale model consists of two stages. In **stage 1**, we will learn the transductive embeddings for all users and items from observed interactions $\mathcal{D}$. In **stage 2**, we will learn the parameters of the two RNNs. In summary, the blue-colored transductive embeddings for active users $\{\phi_u : u \in \mathcal{U}_{act}\}$ and items $\{x_i : i \in \mathcal{I}\}$ are learned in stage 1 and then fixed in stage 2. The red-colored components that include the common initialization $\psi$ for inactive users and the RNN parameters $\Theta_1$ and $\Theta_2$ are learned in stage 2. More specifically,

**Stage 1**. We will first ignore the sequential ordering of the interactions, and learn the transductive user and item embeddings by optimizing the cross-entropy loss (or called softmax loss):

$$\mathcal{L}(\{\phi_u\}, \{x_i\}) = \tfrac{1}{|\mathcal{D}|} \sum_{(u,i) \in \mathcal{D}} \log \left[ \sum_{j \in \mathcal{O}(u,i)} \exp(\phi_u^\top x_j) \right] - \phi_u^\top x_i, \qquad (10)$$

where the set $\mathcal{O}(u, i)$ is the set of items displayed to user $u$ when he/she clicks on item $i$. Typically this information is not given, so we could randomly sample $p$ items as the non-clicked items, and use these $p$ items together with the clicked item $i$ to constitute the offer set $\mathcal{O}(u, i)$.

Stage 1 training is similar to collaborative competitive filtering (CCF) in Yang et al. (2011). It is efficient since the objective is a simple function. Besides, since the objective is convex in either $\phi_u$ or $x_i$, it is easier to obtain the global optimum and the results are affected less by the initialization.

Since active users have many interactions, their transductive embeddings are learned well and lead to good predictions in held-out data. However, for inactive users with fewer observed interactions, the learned transductive embeddings could be overfitted. Thus, in the next stage of training RNNs, we will re-use the transductive embeddings for active users but will discard those for inactive users. Furthermore, the learned item embeddings $\{x_i\}$ will also be used in the next stage of training.

**Stage 2**. We can divide the training set according to active and inactive user as $\mathcal{D}_{act} = \{(u_t, i_t) : (u_t, i_t) \in \mathcal{D} \wedge u_t \in \mathcal{U}_{act}\}$ and $\mathcal{D}_{in} = \{(u_t, i_t) : (u_t, i_t) \in \mathcal{D} \wedge u_t \in \mathcal{U}_{in}\}$. Then we will train the parameters of the two RNNs using the following loss functions:

$$\mathcal{L}_{in}(\Theta_1, \psi) := \tfrac{1}{|\mathcal{D}_{in}|} \sum_{(u_t, i_t) \in \mathcal{D}_{in}} \log \left[ \sum_{j \in \mathcal{O}(u_t, i_t)} \exp(\phi_{u_t}^\top x_j) \right] - \phi_{u_t}^\top x_{i_t}, \qquad (11)$$

$$\mathcal{L}_{act}(\Theta_2) := \tfrac{1}{|\mathcal{D}_{act}^K|} \sum_{(u_t, i_t) \in \mathcal{D}_{act}^K} \log \left[ \sum_{j \in \mathcal{O}(u_t, i_t)} \exp(\phi_{u_t}^\top x_j) \right] - \phi_{u_t}^\top x_{i_t}, \qquad (12)$$

where user embeddings $\{\phi_{u_t}\}$ in Eq. 11 and Eq. 12 are updated sequentially using the corresponding RNNs in Eq. 8 and Eq. 9, respectively. Furthermore, $\mathcal{D}_{act}^K \subset \mathcal{D}_{act} \subset \mathcal{D}$ contains the last $K$ interactions for each active user observed in the training set. That is, for an active user who has more than $t^*$ interactions, we will only use the last $K$ ($K \le t^*$) interaction of the user to train the RNN. First, this allows us to avoid the direct use of RNN for long sequence modeling, which is inefficient. Second, the transductive embeddings for active users have already encoded most information from these users and only a small online adaption is needed to boost the performance further.

Overall, we find that by encoding the history into the transductive embedding and only learn the $K$ step adaptation can largely reduce the computational cost. This reflects another benefit of treating active and inactive using the threshold $t^*$.

## 3.3 Implementation Details

We present two implementation details that are essential for the performance of our model in the experiments but less relevant to the main ideas of this paper.

**2TS-Plus.** We propose a variant of our 2TS model called 2TS-Plus. The only difference from 2TS is that, we replace the user embeddings $\phi_{u_t}$ in 2TS by $\widehat{\phi}_{u_t} \leftarrow W^\top [\phi_{u_t}^\top, x_{i_{t-1}}^\top]^\top$ where $x_{i_{t-1}}$ is the item embedding of the user $u_t$'s most recently clicked item $i_{t-1}$, and $W$ is a learnable weight matrix. In summary, 2TS-Plus explicitly incorporates the information of the 'last clicked item' to compute the user embeddings. Our experiments show that this can consistently improve performance. In fact, we incorporate the 'last clicked item' because we found that the baseline JODIE (Kumar et al., 2019)

performs particularly well on lastfm-1K. Thus we looked into their implementation and found that it is the 'last clicked item' that helps with the performance.

**Features.** In many datasets, item features $\{\boldsymbol{f}_i\}_{i \in \mathcal{I}}$ are provided. In the experiments, we concatenate the transductive item embeddings $\boldsymbol{x}_i$ with the feature-based embedding, $g_\varphi(\boldsymbol{f}_i)$, where $g_\varphi$ a simple network with parameter $\varphi$. The concatenation $[\boldsymbol{x}_i^\top, g_\varphi(\boldsymbol{f}_i)^\top]^\top$ will be used as the item embeddings, and both transductive embeddings $\boldsymbol{x}_i$ and the parameters $\varphi$ are learned in stage 1.

## 4 RELATED WORK

Our stage 1 training is most related to methods based on matrix factorization (MF) (Koren et al., 2009). Many methods for MC are collaborative filtering (CF)-based (Koren et al., 2009). Neural models for recommendation broadly fall into the following three categories.

**Graph-based models.** Monti et al. (2017) first studied recommender systems with graph neural network (GNN). Berg et al. (2017) proposed graph convolutional matrix completion (GCMC) which applies a GNN on the user-item bipartite graph to learn user and item embeddings. It is a transductive model, as it learns user/item-specific embeddings. PinSage (Ying et al., 2018) and IGMC (Zhang & Chen, 2020) are recently proposed inductive GNN-based recommender systems. Despite showing promising results, existing GNN-based recommender systems usually have poor scalability, due to their expensive neighborhood sampling procedure for performing graph convolution.

**Dynamic graph models.** Based on the idea that both users and items will evolve over time via temporal interactions, Dai et al. (2016); Farajtabar et al. (2017); Kumar et al. (2019); Goyal et al. (2020); Sankar et al. (2020) take the graph temporal evolution into modeling design. However, most existing dynamic graph approaches cannot scale to large interaction graphs. JODIE (Kumar et al., 2019) proposed a batching method to make the training process more efficient.

**Deep Sequence models.** RNN (Hidasi et al., 2015; Jannach & Ludewig, 2017) and LSTM (Devooght & Bersini, 2017; Chen et al., 2019) based deep models are widely used in sequential recommendation. Other methods based on attention models (Zhou et al., 2018; 2019) have also been explored. However, these models still have difficulties in leveraging the information contained in states located far into the past due to gradient propagation issues (Pascanu et al., 2013). Several recent advances are proposed to deal with long-range sequences (Tang et al., 2019; Pi et al., 2019)

**Cold-start problem.** Traditional approaches to address cold-start problems include content filtering, hybrid models, etc. Vartak et al. (2017) proposed a meta-learning perspective for the item cold-start problem, where recommending new items to users is formulated as learning a learning algorithm. Lee et al. (2019) proposed to use meta-learning to estimate new user's preferences with a few consumed items. Bose et al. (2019) proposed Meta-Graph to perform few-shot link prediction across multiple graphs. Wu et al. (2020) proposed to compute embeddings for new users using the embeddings of active users, via an attention-based model. They share the same idea of splitting the users into two sets, but their main target is the matrix factorization problem and neither consider the temporal evolution nor the two-time-scale difference.

## 5 EXPERIMENTS

To evaluate the performance of the 2TS model, we conduct experiments on three public datasets, two of which are the largest recommendation benchmark datasets that are closer to industrial scenario, making our results more solid and convincing. By comparing to a diverse set of state-of-the-art (SOTA) methods, we demonstrate the scalability and accuracy of our 2TS model. See detailed configurations in Appendix C.

Table 2: Dataset Statistics

| Dataset | #users | #items | #interactions |
|---------|--------|--------|---------------|
| lastfm-1K | 1,000 | 1,000 | 1,293,103 |
| ML-25M | 162,538 | 59,048 | 24,999,849 |
| Taobao | 987,975 | 4,111,798 | 96,678,667 |

**Dataset.** We consider 3 public datasets: Taobao dataset (Pi et al., 2019), MovieLens-25M (ML-25M) (Harper & Konstan, 2015), and lastfm-1K (Celma, 2010). Dataset statistics are summarized in Table 2. Taobao and ML-25M are large-scale. Especially, Taobao contains about 100 million interactions. The results on these two datasets are much more convincing. All datasets provide user-item interaction data where for each interaction, user ID, item ID, timestamp, and item feature are given. For ML-25M, we ignore the ratings and simply use it as interaction data. For each dataset, we sort the interactions by timestamp, and use the first 70% as the training set, the following 10% as the validation set, and the last 20% as the test set. All reported results are estimated on the test set.

Table 1: Overall performance for interaction prediction. The best method in each column is colored blue and second best is colored light blue . The * symbol indicates that the original implementations released by the authors or the proposed methods are not scalable and will not be able to run on the indicated datasets without our modifications to their methods. The ♯ symbol indicates that the released implementations have been modified by us to better adapt to the dataset and evaluation metric, so that their results in this table should be expected to be better than the original version. The concrete modifications we made are described in Appendix B.

| | method | Taobao | | ML-25M | | lastfm-1K | |
| --- | --- | --- | --- | --- | --- | --- | --- |
| | | MRR | Rec@10 | MRR | Rec@10 | MRR | Rec@10 |
| *CF-based* | CCF♯ | 0.402 | 0.621 | 0.193 | 0.392 | 0.051 | 0.116 |
| *GNN-based models* | GCMC♯ | 0.303 | 0.542 | 0.171 | 0.378 | 0.081 | 0.129 |
| | GCMC-SAGE♯ | 0.149 | 0.366 | 0.109 | 0.264 | 0.168 | 0.208 |
| | GCMC-GAT♯ | 0.230 | 0.494 | 0.185 | 0.404 | 0.064 | 0.118 |
| *Deep sequence models* | SumPooling | 0.415 | 0.664 | 0.302 | 0.591 | 0.071 | 0.180 |
| | GRU4REC♯ | 0.546 | 0.777 | 0.364 | 0.657 | 0.152 | 0.344 |
| | DIEN♯ | 0.605 | 0.834 | 0.356 | 0.638 | 0.100 | 0.213 |
| | MIMN♯ | 0.607 | 0.828 | 0.363 | 0.653 | 0.115 | 0.261 |
| | SASRec♯ | 0.488 | 0.702 | 0.360 | 0.653 | 0.147 | 0.323 |
| | Bert4Rec | 0.280* | 0.480* | 0.397 | 0.652 | 0.216 | 0.369 |
| *Dynamic graph models* | dynAERNN | -oom- | -oom- | 0.249 | 0.509 | 0.021 | 0.038 |
| | JODIE♯ | 0.454* | 0.680* | 0.354* | 0.634* | 0.176 | 0.325 |
| *Our method* | **2TS** | 0.669 | 0.844 | 0.404 | 0.693 | 0.151 | 0.332 |
| | **2TS-Plus** | 0.680 | 0.844 | 0.409 | 0.691 | 0.203 | 0.369 |
| *Improvement to best baseline* | **2TS** | 10.2% | 1.2% | 1.8% | 5.5% | - | - |
| | **2TS-Plus** | 12.0% | 1.2% | 3.0% | 5.2% | - | - |

**Baselines.** We compare 2TS with 12 models spanning 4 categories: (i) **CF-based methods**: Collaborative competitive filtering (CCF) (Yang et al., 2011) is an advanced CF model which takes into account the context of user's choice. It is a simple yet very effective method. (ii) **GNN-based models**: GCMC (Berg et al., 2017) is a SOTA graph-based architecture for recommendation. GCMC-SAGE is its stochastic variant which is more efficient. GCMC-GAT is another variant based on graph attention networks (Veličković et al., 2017). (iii) **Deep sequence models**: SumPooling is a simple yet effective baseline that is widely used in industry. GRU4REC (Hidasi et al., 2015) is a representative of RNN-based models. DIEN (Zhou et al., 2019) is an advanced attention-based sequence model. MIMN (Pi et al., 2019) is a memory-enhanced RNN architecture to better capture long sequential user behaviors. It is a strong baseline and the authors come from the team that publicized Taobao dataset. SASRec (Kang & McAuley, 2018) is a 2-layer transformer decoder-like model. Also built on top of transformer architecture, Bert4Rec (Sun

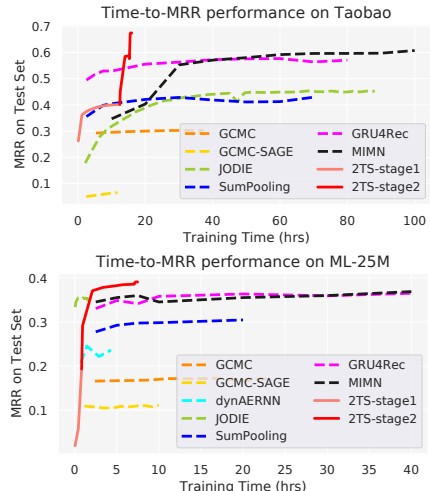

Figure 2: Test MRR versus training time. We compare the convergence speeds of different models in wall-clock time. Run-time on lastfm-1K is not showed because it is small-scale.

et al., 2019) further introduced a BERT type Cloze task with bidirectional attention. (iv) **Dynamic graph models**: JODIE (Kumar et al., 2019) and dynAERNN (Goyal et al., 2020) are two dynamic temporal graph approaches which learn graph node dynamic embeddings via temporal modeling.

**Evaluation Metric.** We measure the performance of different models in terms of the mean reciprocal rank (MRR), the average of the reciprocal rank, and recall@10, which is the fraction of interactions in which the ground truth item is ranked in the top 10. For both metrics, higher values are better. For every interaction, the ranking of ground truth item is calculated with respect to 500 items where the other 499 negative items are randomly sampled from the set of all items.

**Overall Performance** (Table 1). We summarize the overall performance in terms of MRR and Rec@10 in Table 1. On the large-scale datasets Taobao and ML-25M, our models 2TS and 2TS-Plus have consistent and significant improvements compared to all baselines. In terms of MRR, an improvement of 12.0% and 3.0% are achieved by 2TS-Plus, respectively. Note that the lastfm-1K dataset is *much smaller* than Taobao and ML-25M. We run experiments on lastfm-1K just to show that the models have very different behaviors on datasets of different scales. It is interesting to see that, for example, GCMC-SAGE, which performs the worst on both two large-scale datasets, achieves the 3rd best performance on lastfm-1K. The real advantages of our models are for large-scale problems where candidate items are sparse.

**Scalability & Efficiency** (Fig. 2). To compare the training efficiency, we evaluate different models' intermediate checkpoint performances on the test set. Fig. 2 shows that the performance of 2TS increases fast. 2TS's first stage transductive training is efficient and provides an effective initialization for the second stage inductive training. Besides, though not revealed from the figures, the training time of 2TS for each epoch is much smaller than the baselines.

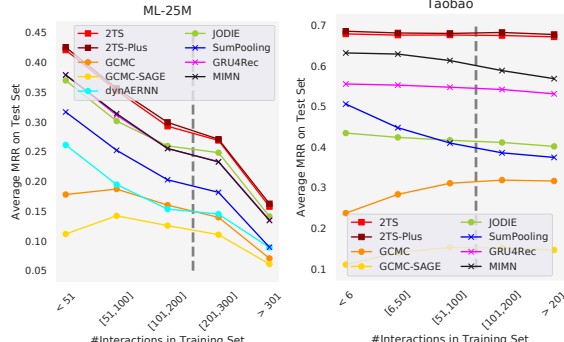

Figure 3: Test MRR in different segments of users.

**Performance for different users.** Fig. 3 shows the MRR performances averaged over users with different numbers of observed interactions (the more interactions the more active the user is). The dash line indicates the threshold that we use to split users into inactive and active groups. 2TS and 2TS-Plus lead to consistent improvements over the entire range of users. On ML-25M, the improvement on inactive users are more obvious, while on Taobao the improvement on active ones is more obvious.

**Two-time-scale behavior** (Fig. 4). Recall 2TS has two RNNs. One updates the embeddings of inactive users and the other updates those of active users. Given the learned RNNs, on the test set, we compute the changes of user embeddings in 2-norm as the user interacts with more items (Fig. 4). The behavior aligns with our intuition. Embeddings of inactive users are changed faster and those of active users are more static.

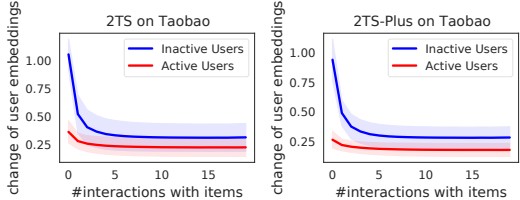

Figure 4: Two-time-scale behavior of user embeddings.

**Ablation Study** (Table 3). To show the effectiveness of the two-stage training, we compare 2TS to 2TS-SingleStage, which has exactly the same architecture as 2TS but all parameters are trained together in a single stage. 2TS-SingleStage performs worse, which means the transductive embeddings are learned very well in Stage 1, benefit from the convexity and easiness of optimization. Besides, the training of 2TS-SingleStage is a lot slower. Furthermore, to show the effectiveness of the 2TS model design, we compare it to 2-GRU, which applies two different RNNs to inactive and active users, but there are no transductive user embeddings for active users.

Table 3: Ablation study.

| method | ML-25M | | Taobao | |
|---|---|---|---|---|
| | MRR | Rec@10 | MRR | Rec@10 |
| **2TS** | 0.404 | 0.693 | 0.669 | 0.894 |
| 2TS-SingleStage | 0.373 | 0.636 | 0.420 | 0.533 |
| 2-GRU | 0.354 | 0.643 | 0.597 | 0.791 |

## 6 CONCLUSION

We proposed to learn two-time-scale representation (2TS) for large recommendation systems to explicitly address the learning efficiency for different user dynamics. We also proposed a two-stage training schema, to leverage the transductive embedding as the inductive model initialization for active users. We evaluated 2TS on large scale recommendation ranking tasks, and showed that 2TS is able to outperform several class of state-of-the-art methods including deep sequence models, graph neural network, and dynamic graphs. Our results show that, designing different representation to capture diverse interaction dynamics are desirable. Future work includes separating and updating user dynamics in an algorithmic way.

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

## A    THEORETICAL MOTIVATION: WHY TWO-TIME-SCALE MODELS?

Here we will provide more details for Section 2. First, we summarize the setting in Section 2.1 formally and mathematically:

**Setting (A).**

- Each *item* is represented by a vector $\boldsymbol{x}$ in a given bounded space $\mathcal{X} := \{\boldsymbol{x} \in \mathbb{R}^d : \|\boldsymbol{x}\|_2 \le B_x\}$.
- Each *user* is represented by a distribution $\mu$ over the space $\mathcal{X} \times \{0,1\}$. Each sample $z := (\boldsymbol{x}, y) \sim \mu$ from this distribution is an item $\boldsymbol{x}$ and a binary label $y$ indicating whether this item is clicked by the user. Further, we assume that a user $\mu$ is drawn from a meta-distribution $\mu \sim p_u$.
- We parameterize the user $\mu$ through a logistic regression model so that a groundtruth user embedding is defined as the minimizer to the expected risk:

$$\boldsymbol{\phi}_\mu^* := \arg\min_{\boldsymbol{\phi} \in \Phi} \mathcal{R}_\mu(\boldsymbol{\phi}), \quad \text{where } \mathcal{R}_\mu(\boldsymbol{\phi}) := \mathbb{E}_{(\boldsymbol{x},y) \sim \mu} \ell(\boldsymbol{\phi}, \boldsymbol{x}, y),$$
$$\text{and } \ell(\boldsymbol{\phi}, \boldsymbol{x}, y) := -y\boldsymbol{x}^\top \boldsymbol{\phi} + \log(1 + \exp(\boldsymbol{x}^\top \boldsymbol{\phi})) + \tfrac{c}{2}\|\boldsymbol{\phi}\|_2^2,$$

  where $c > 0$ is a regularization constant. Assume the parameter space is also bounded $\Phi := \{\boldsymbol{\phi} \in \mathbb{R}^d : \|\boldsymbol{\phi}\|_2 \le B_\phi\}$.
- **Training samples.** Each user is associated with $T$ observed interactions, denoted by $z_{[T]} := \{(\boldsymbol{x}_1, y_1), \cdots, (\boldsymbol{x}_T, y_T)\} \sim \mu^T$. Assume $T \ge 0$ is a random variable that follows a power law distribution with density $p(T) \propto (T+1)^{-\alpha}$, and denote $T \sim p_T^\alpha$. $T$ is independent of $\mu$. Given the observations $z_{[T]}$ for a user $\mu$, an estimate of the user embedding $\phi(z_{[T]})$ can be computed using learning algorithms such as stochastic gradient descent (SGD). In this paper, we assume $\phi(z_{[T]})$ is estimated by SGD with initialization $\phi_0 = \boldsymbol{0}$ and step size $\theta = \frac{1}{c}$ on the loss $\ell$.
- **Test scenario.** Each user $\mu$ can be viewed as an online learning task, where items arrive sequentially. Starting from an initial embedding $\phi_\mu^1$, an online algorithm updates the user embedding whenever it observes a new data point $(\boldsymbol{x}_t, y_t) \sim \mu$:

$$\boldsymbol{\phi}_\mu^{t+1} \leftarrow \text{Update}(\boldsymbol{\phi}_\mu^t, \boldsymbol{x}_t, y_t),$$

  then applies the most updated embedding to $\phi_\mu^{t+1}$ the next item $\boldsymbol{x}_{t+1}$, and suffers from the loss $\ell(\boldsymbol{\phi}_\mu^{t+1}, \boldsymbol{x}_{t+1}, y_{t+1})$. The *excessive risk* of this online algorithm applying to $N$ test samples is

$$\tfrac{1}{N} \sum_{t=1}^N \ell(\boldsymbol{\phi}_\mu^t, \boldsymbol{x}_t, y_t) - \ell(\boldsymbol{\phi}_\mu^*, \boldsymbol{x}_t, y_t).$$

Before the proof of the main theorem, we first introduce the result of stochastic analysis by (Nemirovski et al., 2009), which can be adapted to derive the following proposition that tells the error of the estimation $\phi(z_{[T]})$ from the training samples.

**Proposition A.1.** *Assume Setting (A). Given the $T$ observations $z_{[T]} \sim \mu^T$, if we estimate the user embedding using projected stochastic gradient descent (SGD) with initialization $\boldsymbol{0}$, step size $\frac{1}{c}$, and operated on $\ell_2$-regularized logistic regression loss $\frac{1}{T}\sum_{i=1}^T -y_i \boldsymbol{x}_i^\top \boldsymbol{\phi} + \log(1+\exp(\boldsymbol{x}_i^\top \boldsymbol{\phi})) + \frac{c}{2}\|\boldsymbol{\phi}\|_2^2$, then the expected squared error of the estimation $\phi(z_{[T]})$ can be bounded by*

$$\mathbb{E}_{z_{[T]} \sim \mu^T} \|\phi(z_{[T]}) - \boldsymbol{\phi}_\mu^*\|_2^2 \le \frac{Q(\mu)}{T+1}, \text{ where } Q(\mu) = \max\{\frac{(B_x + cB_\phi)^2}{c^2}, \|\boldsymbol{\phi}_\mu^*\|_2^2\}.$$

*Proof.* Consider the loss $\ell(\boldsymbol{\phi}, \boldsymbol{x}, y) = -y\boldsymbol{x}^\top \boldsymbol{\phi} + \log(1+\exp(\boldsymbol{x}^\top \boldsymbol{\phi})) + \frac{c}{2}\|\boldsymbol{\phi}\|_2^2$. It is easy to verify that $\ell$ is $c$-strongly convex, and that the second moment of the gradient is bounded by

$$\mathbb{E}\|\partial_\phi \ell\|_2^2 = \mathbb{E}\|(\frac{\exp(\boldsymbol{x}^\top \boldsymbol{\phi})}{1+\exp(\boldsymbol{x}^\top \boldsymbol{\phi})} - y)\boldsymbol{x} + c\boldsymbol{\phi}\|_2^2 \le \mathbb{E}(\|\boldsymbol{x}\|_2 + c\|\boldsymbol{\phi}\|_2)^2 \le (B_x + cB_\phi)^2.$$

Then applying the result by (Nemirovski et al., 2009) (more specifically, Equation (2.9) in the paper), we can obtain the above error bound. □

### A.1    PROOF OF THEOREM 2.1.

We first restate Theorem 2.1 as the following theorem, which provides more details. Then we present the proof.

**Theorem A.1** (Detailed version of Theorem 2.1). *Assume Setting (A). Assume the online algorithm for updating the embeddings $\phi_\mu^t$ in Eq. 3 is online gradient descent (OGD) with step size $\gamma$, i.e., $\phi_\mu^{t+1} = \text{Proj}_\Phi \left[ \phi_\mu^t - \gamma \partial \ell(\phi_\mu^t, \boldsymbol{x}_t, y_t) \right]$. Let $z_{[T]}$ denote the observed training samples and $z_{[N]}$ denote the sequence of test samples for the online learning task. With the initialization strategy in Eq. 4, the expected excessive risk is bounded by*

$$\mathbb{E}_{(\mu,T)\sim(p_u\times p_T^\alpha)}\mathbb{E}_{z_{[T]}\sim\mu^T}\mathbb{E}_{z_{[N]}\sim\mu^N}\frac{1}{N}\sum_{t=1}^N\ell(\phi_\mu^t,\boldsymbol{x}_t,y_t)-\ell(\phi_\mu^*,\boldsymbol{x}_t,y_t)$$

$$\leq \frac{\beta(t^*)+\gamma^2(B_x+cB_\phi)^2N}{2\gamma N}, \tag{13}$$

*where*

$$\beta(t^*) = \frac{\zeta(\alpha+1,t^*+1)}{\zeta(\alpha,1)}\bar{Q} + \left(1 - \frac{\zeta(\alpha,t^*+1)}{\zeta(\alpha,1)}\right)\text{Var}_{\boldsymbol{m}}$$

$$\bar{Q} = \mathbb{E}_{\mu\sim p_u}Q(\mu) = \mathbb{E}_{\mu\sim p_u}\max\{\frac{(B_x+cB_\phi)^2}{c^2},\|\phi_\mu^*\|_2^2\},$$

$$\text{and } \zeta(a,b) = \sum_{i=0}^\infty (i+b)^{-a} \text{ is the Hurwitz zeta function .}$$

*Choosing the optimal step size $\gamma = \frac{1}{B_x+cB_\phi}\sqrt{\frac{\beta(t^*)}{N}}$, the upper bound is*

$$\mathbb{E}\frac{1}{N}\sum_{t=1}^N\ell(\phi_\mu^t,\boldsymbol{x}_t,y_t)-\ell(\phi_\mu^*,\boldsymbol{x}_t,y_t) \leq \frac{1}{B_x+cB_\phi}\sqrt{\frac{\beta(t^*)}{N}}. \tag{14}$$

*Besides, the optimal choice of the threshold $t^*$ is*

$$t^{**} = \underset{t^*\geq 0}{\arg\min}\,\beta(t^*) = \left\lfloor \frac{\bar{Q}}{\text{Var}_{\boldsymbol{m}}} \right\rfloor - 1. \tag{15}$$

*Proof.* Denote the $N$ test samples by $z_{[N]} \sim \mu^N$. Following the analysis and results in (Nemirovski et al., 2009), one can show that

$$\mathbb{E}_{z_{[N]}\sim\mu^N}\|\phi_\mu^{t+1}-\phi_\mu^*\|_2^2 \leq \mathbb{E}_{z_{[N]}\sim\mu^N}\|\phi_\mu^t-\phi_\mu^*\|_2^2$$
$$- 2\gamma\mathbb{E}_{z_{[N]}\sim\mu^N}\left[(\phi_\mu^t-\phi_\mu^*)^\top\partial\ell(\phi_\mu^t,\boldsymbol{x}_t,y_t)\right] + \gamma^2(B_x+cB_\phi)^2.$$

The above inequality corresponds to equation (2.6) in (Nemirovski et al., 2009). Then, since $\ell$ is convex in $\phi$, then

$$\mathbb{E}_{z_{[N]}\sim\mu^N}\left[(\phi_\mu^t-\phi_\mu^*)^\top\partial\ell(\phi_\mu^t,\boldsymbol{x}_t,y_t)\right] \geq \mathbb{E}_{z_{[N]}\sim\mu^N}\left[\ell(\phi_\mu^t,\boldsymbol{x}_t,y_t)-\ell(\phi_\mu^*,\boldsymbol{x}_t,y_t)\right].$$

Denote $e_t = \mathbb{E}_{z_{[N]}\sim\mu^N}\|\phi_\mu^t-\phi_\mu^*\|_2^2$ and combine the above two inequalities, we have

$$2\gamma\mathbb{E}_{z_{[N]}\sim\mu^N}\left[\ell(\phi_\mu^t,\boldsymbol{x}_t,y_t)-\ell(\phi_\mu^*,\boldsymbol{x}_t,y_t)\right] \leq e_t - e_{t+1} + \gamma^2(B_x+cB_\phi)^2.$$

Summing over $t$, we have

$$\mathbb{E}_{z_{[N]}\sim\mu^N}\frac{1}{N}\sum_{t=1}^N\ell(\phi_\mu^t,\boldsymbol{x}_t,y_t)-\ell(\phi_\mu^*,\boldsymbol{x}_t,y_t) \leq \frac{\|\phi_\mu^1-\phi_\mu^*\|_2^2}{2\gamma N} + \frac{\gamma(B_x+cB_\phi)^2}{2}$$

Taking expectation over the training samples $z_{[T]}$, $T$, and $\mu$, we have

$$\mathbb{E}_{(\mu,T)\sim(p_u\times p_T^\alpha)}\mathbb{E}_{z_{[T]}\sim\mu^T}\mathbb{E}_{z_{[N]}\sim\mu^N}\frac{1}{N}\sum_{t=1}^N\ell(\phi_\mu^t,\boldsymbol{x}_t,y_t)-\ell(\phi_\mu^*,\boldsymbol{x}_t,y_t)$$

$$\leq \mathbb{E}_{(\mu,T)\sim(p_u\times p_T^\alpha)}\mathbb{E}_{z_{[T]}\sim\mu^T}\frac{\|\phi(z_{[T]})-\phi_\mu^*\|_2^2}{2\gamma N}\mathbf{1}\,[T\geq t^*]$$

$$+ \mathbb{E}_{(\mu,T)\sim(p_u\times p_T^\alpha)}\frac{\|\boldsymbol{m}-\phi_\mu^*\|_2^2}{2\gamma N}\mathbf{1}\,[T<t^*]$$

$$+ \frac{\gamma(B_x+cB_\phi)^2}{2}.$$

Regarding the first term on the right hand side, by Proposition A.1,

$$\mathbb{E}_{(\mu,T)\sim(p_u\times p_T^\alpha)}\mathbb{E}_{z_{[T]}\sim\mu^T}\|\phi(z_{[T]})-\phi_\mu^*\|_2^2\mathbf{1}\left[T\geq t^*\right]$$

$$\leq \mathbb{E}_{(\mu,T)\sim(p_u\times p_T^\alpha)}\frac{Q(\mu)\mathbf{1}\left[T\geq t^*\right]}{T+1} = \mathbb{E}_{\mu\sim p_u}Q(\mu)\mathbb{E}_{T\sim p_T^\alpha}\frac{\mathbf{1}\left[T\geq t^*\right]}{T+1}$$

$$= \bar{Q}\mathbb{E}_{T\sim p_T^\alpha}\frac{\mathbf{1}\left[T\geq t^*\right]}{T+1} = \bar{Q}\sum_{T=t^*}^{\infty}\frac{1}{T+1}\frac{(T+1)^{-\alpha}}{\zeta(\alpha,1)}$$

$$= \bar{Q}\frac{\zeta(\alpha+1,t^*+1)}{\zeta(\alpha,1)}.$$

Regarding the second term,

$$\mathbb{E}_{(\mu,T)\sim(p_u\times p_T^\alpha)}\|\boldsymbol{m}-\phi_\mu^*\|_2^2\mathbf{1}\left[T<t^*\right]$$

$$= \mathrm{Var}_{\boldsymbol{m}}\mathbb{E}_{T\sim p_T^\alpha}\mathbf{1}\left[T<t^*\right] = \mathrm{Var}_{\boldsymbol{m}}\left(1-\frac{\zeta(\alpha,t^*+1)}{\zeta(\alpha,1)}\right).$$

Combining the above inequalities, we can obtain the first bound in Eq. 13, and Eq. 14 follows. Now we need to find the optimal choice of the threshold $t^*$. Note that

$$\beta(t^*) = \mathbb{E}_{T\sim p_T^\alpha}\left[\frac{\bar{Q}}{T+1}\mathbf{1}\left[T\geq t^*\right] + \mathrm{Var}_{\boldsymbol{m}}\mathbf{1}\left[T<t^*\right]\right] \geq \mathbb{E}_{T\sim p_T^\alpha}\min\left\{\frac{\bar{Q}}{T+1},\mathrm{Var}_{\boldsymbol{m}}\right\}.$$

Since $\frac{\bar{Q}}{T+1}$ monotonely decreases as $T$ increases, it is easy to see that

$$\min\left\{\frac{\bar{Q}}{T+1},\mathrm{Var}_{\boldsymbol{m}}\right\} = \begin{cases} \mathrm{Var}_{\boldsymbol{m}}, & \text{when } T < \frac{\bar{Q}}{\mathrm{Var}_{\boldsymbol{m}}}-1 \\ \frac{\bar{Q}}{T+1}, & \text{when } T, \geq \frac{\bar{Q}}{\mathrm{Var}_{\boldsymbol{m}}}-1. \end{cases}$$

Therefore, the lower bound of $\beta(t^*)$ achieves at

$$t^{**} = \left\lfloor\frac{\bar{Q}}{\mathrm{Var}_{\boldsymbol{m}}}\right\rfloor - 1,$$

and it is the optimizer. □

## A.2 DETAILS FOR IMPLICATION 2 IN SECTION 2

Regarding **Implication 2** in Section 2, here we provide more derivation steps for the optimal step sizes for the two group of users.

The expected excessive risk can be written as the sum of the excessive risks of the two groups of users:

$$\mathbb{E}\left[\frac{1}{N}\sum_{t=1}^N\ell(\phi_\mu^t,\boldsymbol{x}_t,y_t)-\ell(\phi_\mu^*,\boldsymbol{x}_t,y_t)\right]$$

$$= \mathbb{E}\left[\left(\frac{1}{N}\sum_{t=1}^N\ell(\phi_\mu^t,\boldsymbol{x}_t,y_t)-\ell(\phi_\mu^*,\boldsymbol{x}_t,y_t)\right)\mathbf{1}[T<t^*]\right] \quad \text{for inactive users}$$

$$+ \mathbb{E}\left[\left(\frac{1}{N}\sum_{t=1}^N\ell(\phi_\mu^t,\boldsymbol{x}_t,y_t)-\ell(\phi_\mu^*,\boldsymbol{x}_t,y_t)\right)\mathbf{1}[T\geq t^*]\right] \quad \text{for active users}$$

With similar derivation steps as the proof for Theorem A, we can see that for inactive users, the bound is

$$\mathbb{E}\left[\left(\frac{1}{N}\sum_{t=1}^N\ell(\phi_\mu^t,\boldsymbol{x}_t,y_t)-\ell(\phi_\mu^*,\boldsymbol{x}_t,y_t)\right)\mathbf{1}[T<t^*]\right]$$

$$\leq \mathbb{E}\left[\left(\frac{\mathrm{Var}_{\boldsymbol{m}}}{2\gamma N}+\frac{\gamma(B_x+cB_\phi)^2}{2}\right)\mathbf{1}[T<t^*]\right] = \left(\frac{\mathrm{Var}_{\boldsymbol{m}}}{2\gamma N}+\frac{\gamma(B_x+cB_\phi)^2}{2}\right)\Pr[T<t^*].$$

Optimizing the step size $\gamma$ gives $\gamma_{in} = \frac{1}{B_x + cB_\phi} \sqrt{\frac{\text{Var}_m}{N}}$.

Similarly, for active users, the bound is

$$\mathbb{E}\left[\left(\frac{1}{N}\sum_{t=1}^{N}\ell(\phi_\mu^t, \boldsymbol{x}_t, y_t) - \ell(\phi_\mu^*, \boldsymbol{x}_t, y_t)\right)\mathbf{1}[T \geq t^*]\right]$$

$$\leq \mathbb{E}\left[\left(\frac{\bar{Q}}{(T+1)2\gamma N} + \frac{\gamma(B_x + cB_\phi)^2}{2}\right)\mathbf{1}[T \geq t^*]\right]$$

$$= \left(\frac{\bar{Q}\zeta(\alpha+1, t^*+1)}{2\gamma N} + \frac{\gamma(B_x + cB_\phi)^2\zeta(\alpha, t^*+1)}{2}\right)/\zeta(\alpha, t^*+1).$$

Optimizing the step size $\gamma$ gives $\gamma_{act} = \frac{1}{B_x + cB_\phi} \sqrt{\frac{\zeta(\alpha+1, t^*+1)\bar{Q}}{\zeta(\alpha, t^*+1)N}}$.

## B    BASELINE SPECIFICATION

Some of the compared baseline methods are not directly scalable to large scale interaction graphs, or originally designed for ranking task. To make the baselines runnable on large and sparse graphs and comparable to our proposed method, we have made a few adaptions.

- **CCF**: Since item features are provided in the datasets, to allow CCF to make use of the features, we modify it in the same way as how we incorporate features into our 2TS model (see Section 3.3).

- **JODIE**: we made the following adaptions: 1) replaced the static one-hot representation to 64-dim embeddings because the number of nodes are large; 2) represent the category / tag categorical feature via a learnable embedding table; 3) used triplet loss with random negative example rather than original MSE loss, which empirically show improvements.

- **dynAERNN**: These two methods are originally designed for discrete graph snapshots, while our focused tasks are continues interaction graphs. We manually transformed the interactions graph into 10 snapshots, with equal edge count increments.

  For the downstream ranking task, we followed the evaluation method used in dySat (Sankar et al., 2020): after the node embeddings are trained, we train a logistic regression classifier to predict link between a pair of user / item node embedding using Hadmard operator. The logits on test set are used for ranking. For Taobao dataset, we are not able to get reslt of dynAERNN given the memory constraint.

- **GRU4Rec & DIEN & MIMN**: We changed the data batching from per-user based to per-example based, to better adapt for time ordered interaction data and to better make the full use of the training data. Besides, the implementation of GRU4Rec follows implementation by the authors of MIMN paper, which includes some modifications compared to the original version of GRU4Rec released by their authors, and should be expected to perform better.

- **Bert4Rec**: For ML-25M and lastfm-1k dataset, we followed the hyper-parameter setting for ML-20M dataset, according to the implementation released by the authors, and changed dimension to 128. For Taobao dataset, to fit the model size into 16GB GPU memory, we changed the embedding dimension to 32 and batch size to 8.

- **GCMC & GCMC-SAGE & GCMC-GAT**: We changed the loss function from Softmax over different ratings to Softmax over [true item, 10 random selected items], to better adapt to the ranking task.

Besides, we clarify that in the baseline models Bert4Rec, SASRec, and dynAERNN, item features are not incorporated, since it's not straightforward to include features into their original model design, and it's unclear if adding features will help their setup or not.

## C    CONFIGURATION OF 2TS AND 2TS-PLUS

We will present some important hyperparameters chosen for the 2TS and 2TS-Plus in the experiments.

- Threshold $t^* \in \mathbb{N}$: Users with more than $t^*$ interactions observed in the training set are considered active users, and those with less than or equal to $t^*$ observed interactions are considered inactive users. For the results that we get in Table 1, the choices of the thresholds are given in Table 4.

Table 4: For the results shown in Table 1, the thresholds $t^*$ for splitting the users are specified as follows.

|          | Taobao | Movielens-25M | lastfm-1K |
|----------|--------|---------------|-----------|
| 2TS      | 100    | 200           | 100       |
| 2TS-Plus | 100    | 200           | 200       |

- Last $K$ interactions for active users: In Stage 2 training, for each active user, we only use the last $K$ interactions to train the RNN. The number $K$ is chosen to be either 10 or 20, and they actually give similar performances.
- Embedding dimension: Both user embeddings and item embeddings are set to be of dimension 64. If the item contains features, we will use another 64 dimension for the feature embedding.
- Learning rate: the learning rate for training 2TS and 2TS-Plus is searched over $1e-3$ and $1e-4$ only. Most of the time $1e-4$ works better. The optimizer is Adam.

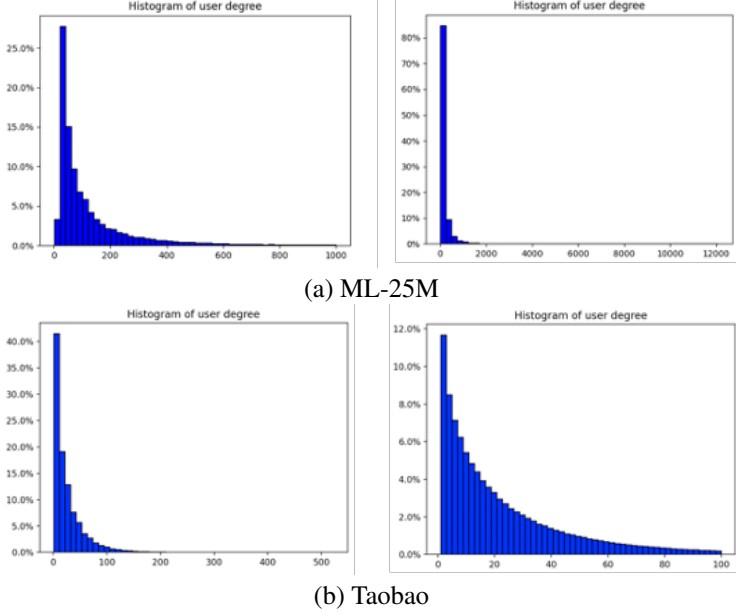

(a) ML-25M

(b) Taobao

Figure 5: Distribution of the number of observed interactions in the training dataset.

