# OpenReview forum: "Learning Two-Time-Scale Representations For Large Scale Recommendations"
_ICLR.cc/2021/Conference — Reject_

### Official Review · AnonReviewer1 · 2020-10-27

**Rating:** 3
**Confidence:** 5

**Review:**

The paper considers the sequential recommendation problem. The proposed method essentially combines the following two ideas: (i) two-stage learning: using conventional CF to pretrain user/item embeddings, and feed them (fixed, unlearned) into the 2nd stage learning. (ii) two-time-scale: using 2 RNNs to model active users and inactive users respectively.

Although the model design makes sense, it seems too ad-hoc and not novel. For example, the idea (i) actually learn the model with pretrained embeddings, which achieves faster training speed. But this has been widely adopted in various domains. The comparing baselines certainly could easily adapt to use the same pretrained embeddings for faster speed. The claimed advantages are not convincing given important related work/baselines are missing. Specifically I have the following concerns:

- Encoding long-range sequence: the proposed method doesn't model long sequences, while seeks to capture user history with a simple MF. This idea: (i) ignores sequential patterns in user previous history; (ii) has been widely adopted in the literature[1][2], especially before we have powerful deep sequential models.

- Computational efficiency: (i) the efficiency advantage mostly comes from the use of pretrained embeddings, which can be adopted by almost all the baselines for acceleration. (ii) Another trick used is to cut off the sequence (only train the last 10/20 user actions). Such an acceleration, again, could be adopted by other methods.

- Baselines: The paper claims SOTA performance on sequential recommendation, but baselines like SASRec(2018), BERT4Rec(2019) are missing. SASRec performed better than GRU4Rec and also showed much faster training speed. The GRU4REC is almost the 2nd best method among all the baselines used in the paper, and it's from 2015.

- table 1 may have some typo. The recall of 2TS on lastfm-1K is worse than GRU4Rec, but the improvement shows 2.2%.

The proposed mode design, though, makes sense, but lacks novelty as it's a combination of various commonly used tricks. The improved training speed comes from pretrained emb and sequence cut-off, which are not a novel contribution for efficient sequential/recommendation model training. Several strong baselines are missing, which further weaken the paper.

========================brief update after author response==============================
I decided to maintain my rating due to several key claims in the response are unverified. For example, in Q1 "We propose that NOT ALL users (but only active users) should use transductive embeddings to memorize their history, which is different from existing methods that treat all users in the same way.", however, comparison against them is missing, especially for methods like [1] and [3] that uses both inductive and sequential embeddings; in Q2 "Trivially cutting off the sequence could result in worse accuracy", however, this is not verified: not clear what's the effect of cutting off on baseline models. I believe these are the core research questions that need to be verified in the paper, which are unfortunately missing.

[1]Fusing Similarity Models with Markov Chains for Sparse Sequential Recommendation, ICDM'16

[2]Personalized Top-N Sequential Recommendation via Convolutional Sequence Embedding, WSDM'18

[3] Next Item Recommendation with Self-Attention

---

> ### Author Response · Authors · 2020-11-24
> **Author Response To Reviewer 1**
>
> We would like to thank Reviewer 1 for the detailed comments! We respond to the concerns as follows:
>
> ***Q1. “...capture user history with a simple MF. This idea ...has been widely adopted.” ***
>
> As Reviewer 1 mentioned,  some existing works have used transductive embeddings to memorize long history and we agree with that. However, we would like to emphasize that, in our paper, the key aspect that existing methods do not consider is the differential use of transductive and inductive embeddings for different groups of users. That is,
>
> - We propose that NOT ALL users (but only active users) should use transductive embeddings to memorize their history, which is different from existing methods that treat all users in the same way.
>
> - For inactive users with few observations, we propose to use an inductive embedding that is shared across them. More importantly, we show that for inactive users this is better than using transductive embeddings to encode history. In the revised paper, we include a theoretical analysis to rigorously justify why we need to split the users into different groups and the benefits of differential use of transductive and inductive embeddings for the two groups.
>
> To conclude, our work is novel compared to existing ones because it treats different groups of users differently, explains why we should do so, and also demonstrates the improvements experimentally. Both Reviewer 3 and 4 think our idea is interesting.
>
> Talking about the novel contribution of our work, we will be very appreciative if Reviewer 1 could kindly read the newly added theoretical analysis (as suggested by Reviewer 2) in Section 2, where we rigorously explain (i) why a differential use of transductive and inductive embeddings for active and inactive users is expected to lead to better performance, and (ii) why the embeddings for active and inactive users should evolve in different time scales. We sincerely hope this could be considered as an additional contribution when our work is evaluated.
>
> ***Q2. “cut off the sequence … again, could be adopted by other methods.”***
>
> We would like to argue that it is not so trivial to reduce the computational cost and at the same time achieve high accuracy. We can cut off the sequence to save computation only because we adopt transductive embeddings for active users which are already good enough for representing these users. This is why only a very few steps of adaptations are needed for these users and we can cut off the sequences to about 10/20 items during the training, while for inactive users we still use their full history. Trivially cutting off the sequence could result in worse accuracy.
>
> ***Q3. “baselines like SASRec(2018), BERT4Rec(2019) are missing... The GRU4REC is from 2015” ***
>
> - Thank Reviewer 1 for pointing out two additional baselines. We have run both SASRec and BERT4Rec and their performances are now presented in Table 1 in the revised paper.
>
> - Both SASRec and BERT4Rec perform badly on Taobao, which contains millions of items. BERT4Rec did perform well on lastfm-1K, but it is much smaller than Taobao and Movielens-25M (kindly refer to Table 2). It seems these two methods can only perform well when there are fewer items, which is not the case for real-world problems.
>
> - In fact, BERT4Rec is not runnable on Taobao (oom issue) because there are 4M items. We modify their released implementation specifically for BERT4Rec on Taobao and obtain the numbers in Table 1.
>
> - Many methods, though published recently, may not be very convincing since they do not conduct experiments on large-scale datasets in their paper (that includes SASRec and also several dynamic graph models that we use as the baselines). Therefore, we consider MIMN and DIEN as very strong baselines because they are both released recently and also have conducted experiments on industrial-level datasets. They both perform the best among the baselines on the largest public dataset: Taobao. From our experiments, we could see that the methods behave very differently on different datasets. For example, Bert4Rec, JODIE, GCMC-SAGE perform well on lastfm-1k which is small, but perform very badly on Taobao. Our method has convincing performances on large-scale datasets, which are our main target.
>
> - Finally, we would like to claim that we have put a decent amount of effort into reproducing and even improving the performances of the baselines for a fair comparison. In Table 1, as indicated by the symbol *, some implementations released by the authors or their proposed methods are not scalable and unable to run on large-scale datasets without our modifications to their methods/codes. As indicated by the symbol #, some released implementations have been modified by us to better adapt to the dataset and evaluation protocol, so that their results in Table 1 should be expected to be better than their original version.
>
>
> *** Typo***
>
> Thanks for pointing out the typos! We have gone through the paper to correct them.

---

### Official Review · AnonReviewer3 · 2020-10-29
**Interesting idea, unclear if effective**

**Rating:** 6
**Confidence:** 4

**Review:**

The main idea in this work is to separately model users with a lot of activity in the systems and users with little activity and data.
To this end, two RNN's are trained on the items that the users have interacted with one for active users and one for less active users. User and item embeddings are generated with a matrix factorization process and these are used as input to the RNN's and to initialize its hidden state. For inactive users, a common initialization is used.

While this is an interesting idea it is somewhat unclear what the real benefits of this method are. The split between active and inactive users is rather unclear, how many interactions are then left in the inactive users split?
The experimental results are also somewhat unclear. it seems that the last interacted item is concatenated to the user profile and item attributes are also concatenated to the item representations learned by matrix factorization. Are there additional data sources included in the other baselines as input somehow? It is also unclear if the improved results are due to inactive or active users.

I also think that it would be interesting if the two models would be linked somehow explicitly rather than 'just' from the joint factorization. Overall an interesting idea but the experimental results I think are not convincing and the split modeling could be justified quantitavly.

minor: In table 1 last column the second-best performing method is GRU4Rec.
              Figure 2, is not very informative for large timeframes as most methods converge in a few hours (or minutes) on this data


After reading rebuttal, I think the paper is somewhat improved hence I increase the score.

---

> ### Author Response · Authors · 2020-11-24
> **Author Response To Reviewer 3**
>
> We would like to thank Reviewer 3 for the positive comments on our idea and the questions to the experiments! In the revised paper, we’ve included all the important details suggested by Reviewer 3 to make everything more clear and to avoid confusion. Regarding the questions raised by Reviewer 3:
>
>
> ***Q1. “it seems that the last interacted item is concatenated to the user profile and item attributes are also concatenated ... Are there additional data sources included in the other baselines as input somehow?”***
>
> - For the item attributes (i.e., item features), most baselines have made use of them. Only dynAernn and the newly added baselines (Bert4Rec and SASRec) do not use them, because they are not designed for incorporating features and it’s really not straightforward to include features into their original model design, and it’s unclear if adding features will help their setup or not. In the revised version we’ve clarified this to avoid confusion.
>
> - Furthermore, CCF is also not designed for incorporating features, but we modify it to allow it to make use of the features since the modification is not that difficult. For JODIE, we improve its way of using the item features to obtain even better performance than its original implementation.
>
> - The concatenation of the last interacted item does not mean that we are using more information or more data. Other sequence models in the baselines are also aware of the last interacted item (and also the order of all interactions). The concatenation of the last item should be viewed as the architecture design of 2TS-Plus, which is also adopted by the baseline JODIE. Besides, the 2TS version of our model does not concatenate the last interacted item, and it still performs better than all baselines on large-scale datasets.
>
> - Finally, we would like to claim that we have put a significant amount of effort into reproducing and even improving the performances of the baselines for a fair comparison. In Table 1, as indicated by the symbol *, some implementations released by the authors or their proposed methods are not scalable and unable to run on large-scale datasets without our modifications to their methods/codes. As indicated by the symbol #, some released implementations have been modified by us to better adapt to the dataset and evaluation protocol, so that their results in Table 1 should be expected to be better than their original version.
>
>
> ***Q2. “The split between active and inactive users is rather unclear, how many interactions are then left in the inactive users split? ...  the split modeling could be justified quantitatively.”***
>
> - Thank Reviewer 3 for pointing out the unclarity issue. We have now included Table 4 in Appendix C to clearly specify the splitting thresholds that we used to get the results shown in Table 1. The splitting threshold is 100 on Taobao dataset and is 200 on Movielens-25M, for both 2TS and 2TS-Plus.
>
> - Besides, in the revised paper, we include theoretical analysis in Section 2 to show the benefits of splitting users into active and inactive groups. In the newly provided theorem, Eq 6 quantitatively shows the optimal choice of the splitting threshold. Though the theorem is based on a more simplified setting than the real-world recommendation problems, the results shown by Eq 6 can suggest that the spitting threshold could be chosen to be larger if the variance among the users is smaller.
>
> - Talking about the additional theoretical analysis (as suggested by Reviewer 2) in Section 2, it will be very appreciated if Reviewer 3 could also kindly read Section 2 (proofs are given in Appendix A) and reconsider this as another significant contribution of this paper. Briefly, we rigorously explain: (i) why the embeddings for active and inactive users should evolve in different time scales and two online learning (or adaptation) algorithms are needed. (ii) why a differential use of transductive and inductive embeddings for active and inactive users is expected to lead to better performance.
>
>
> ***Q3. “It is also unclear if the improved results are due to inactive or active users”***
>
> - We've updated the plots in Fig 3 to make them clearer. Fig 3 shows the performances on users grouped by different numbers of observed interactions. It shows that 2TS performs better than the candidates for both inactive and active users. On ML-25M, the improvement in inactive users is more obvious. On Taobao, the improvement in active ones is more obvious.
>
>
> *** Typos and figures***
>
> We have gone through the paper to correct all the typos. Regarding the figures, we have re-plotted Figure 2 to focus on a shorter time training comparison as suggested by the reviewer. We have also changed the color scheme of Figure 2 and 3 to make them clearer. Please let us know whether they still look unclear and we can further improve them.

---

### Official Review · AnonReviewer4 · 2020-10-30
**Interesting Approach towards solving for real-world challenges in RecSys**

**Rating:** 7
**Confidence:** 4

**Review:**

Authors propose an interesting application of Recurrent Neural Networks to build a sequential recommender system with a different model for active and inactive users.

In a real-world recommender system, handling cold-start users separately is a big challenge, and this paper proposes an interesting approach to solve that.

Authors propose a two-staged method, wherein the first stage, the aggregated user-item interaction matrix is factorized to generate initial transductive embeddings for users and items. These are later re-used in the second stage and for active users, their embeddings are fixed.

Some strong points of the paper:

1) A modular approach is taken to handle active users and inactive users separately. This would be helpful in designing real-world recommender systems, where the cold start problem is critical.

2) The paper is generally well-written with an adequate description of the method and any equation.

3) The solution is well-motivated with proper justification in Section 3 and in the introduction.

4) Performs well against strong recent methods like JODIE (Kumar et al) and GNNs.

Some questions to authors:

1) How are cold-start items handled in the method? I assume cold-start items is an equally critical problem faced in designing a RecSys system.

2) Why are item embeddings kept constant in the second stage? For frequent items, learning/updating embeddings during RNN training would enhance their quality.

---

> ### Author Response · Authors · 2020-11-24
> **Author Response To Reviewer 4**
>
> We would like to thank Reviewer 4 for the overall positive comments! Regarding the two questions raised by Reviewer 4:
>
> ***Q1.  How are cold-start items handled in the method? I assume cold-start items are an equally critical problem faced in designing a RecSys system. ***
>
> We have also thought about exactly the same question! Similar to the users that interact with items sequentially, items also interact with users sequentially, e.g. clicked or not. Therefore, symmetrically, an item could also be represented by the sequence of its interacted users, and we can use an RNN to aggregate the sequence of interacted users to compute the item embedding. Considering the number of times visited by the users, we could also divide the items into two groups: *popular items* and *unpopular items* (or cold-start items). Then popular items could be represented by transductive embeddings, and unpopular items could be represented by a common initialization and an RNN which takes the interacted users as the inputs. The way of computing the embeddings of unpopular items could be exactly the same as how we compute the embeddings for inactive users.
>
> Our current paper proposes the two-time-scale representation for *user sequence models*. If in addition, the item side is symmetrically considered, where items are also divided into popular and unpopular groups, then we will have a two-time-scale representation for *dynamic graph models*. Since the implementation for including the item side is nontrivial, we consider this as future work. However, our intuition, design, and theoretical analysis for the user model could also be applied to the item side.
>
> ***Q2. Why are item embeddings kept constant in the second stage? For frequent items, learning/updating embeddings during RNN training would enhance their quality. ***
>
> One reason for keeping the item embeddings constant in the 2nd stage is to improve the computational efficiency. We found that including the item embeddings as learnable parameters together with the RNN learning will significantly increase the memory requirement and also the computation complexity, which is not suitable for large-scale datasets such as Taobao and Movielens-25M. The second reason is that, according to our experimental results, the item embeddings learned in the 1st stage are already good enough and can lead to state-of-the-art performances (Table 1). Therefore, in our current implementation, to save computation, we did not update the item embeddings during RNN training, although a further improvement is expected if we do so.
>
>
> Finally, we would like to thank the comments from Reviewer 4 again, and we would also like to draw Reviewer 4’s attention to two additional contributions that we made in the revised paper:
> - We include theoretical analysis in Section 2 and the proofs in Appendix A to rigorously explain: (i) why the embeddings for active and inactive users should evolve in different time scales and two online learning (or adaptation) algorithms are needed. (ii) why a differential use of transductive and inductive embeddings for active and inactive users is expected to lead to better performance.
> It will be very appreciated if reviewers could read Section 2 and reconsider this as another significant contribution of this paper.
> - We include three additional baselines (GCMC-GAT, SAR4Rec, and Bert4Rec) in Table 1. Two of them (SAR4Rec and Bert4Rec) are suggested by Reviewer 1. Our method shows clear advantages on large-scale datasets (Note that lastfm-1K is much much smaller than Taobao and Movielens-25M).

---

### Official Review · AnonReviewer2 · 2020-10-30
**A reasonable work.**

**Rating:** 6
**Confidence:** 5

**Review:**

Given sequential interactions between users and items, the goal is to predict the next interactions of users.

Challenges
- For active users, common models such as RNNs have limitations when dealing with long-range sequences due to the difficulty in gradient propagation.
- For inactive users (or new users), there exists the cold-start problem.

Main idea
- They partition the users into two sets: active and inactive users. Active users have many observed interactions; Inactive users have very few or zero observed interactions.
- They pre-train user and item embedding vectors via the matrix factorization.
- Two independent RNNs are trained for the active and the inactive users, respectively.
- When training the RNNs, user and item embedding vectors are fixed.
- For all the inactive users, a common embedding initialization is shared.

Strengths
- They tried to explain the intuition of the proposed model in detail.
- The proposed method outperforms some previous methods on real-world datasets.
- The ablation study shows the effectiveness of the proposed training algorithm.

Weaknesses
- The idea is very incremental compared to previous methods.
- There is no theoretical analysis of the proposed method.
- The presentation of the figures is not good (Colors in Figures 2 and 3).

Summary
- They suggested the need to separate learning for active and inactive users. The intuition behind the method makes sense, and it has been shown through experiments that it performs well.
- Authors will be able to increase their contribution through thorough theoretical analysis or developed ideas.
- Authors also need to improve the presentation of the paper.

---

> ### Author Response · Authors · 2020-11-24
> **Author Response To Reviewer 2**
>
> We thank Reviewer 2 for the very constructive suggestion! Based on the suggestions we have revised our paper, and here are our responses to the raised concerns:
>
> ***Q1. There is no theoretical analysis of the proposed method ***
>
> To better motivate the proposed 2TS model, as suggested, we provide a theoretical analysis of the benefits of the 2TS model in the revised version. The analysis is presented in Section 2 and the proofs are given in Appendix A. It will be very appreciated if reviewers could read Section 2 and reconsider this as another significant contribution of this paper. Here is a brief summary:
>
> Through the lens of online learning and stochastic optimization, our analysis is carried out in a learning-to-learn setting for designing online learning algorithms. In this setting, the estimation of the embedding vector of each user is considered as an online learning task, and we assume that the variance of user embeddings is small. Based on this setting, we can theoretically explain:
> (i) why the embeddings for active and inactive users should evolve in different time scales, and two online learning (or adaptation) algorithms are beneficial for better modeling the users;
> (ii) why a differential use of transductive and inductive embeddings for active and inactive users is expected to lead to better performance.
>
> Although the setting for the analysis can not cover all aspects of real-world recommendation problems, it leads to clear insights on the 2TS behavior of active and inactive users, and the benefits of using two different RNNs for these two user groups.
>
> ***Q2. The presentation of the figures is not good (Colors in Figures 2 and 3) ***
>
> Regarding the presentation of the figures, we have replotted Figure 2 and 3, and have also polished Figure 1. Please let us know whether they still look unclear and we can further improve them.

---

### Author Response · Authors · 2020-11-24
**Summary of major revisions**

We would like to thank all the reviewers for their careful reading and constructive comments! We have responded to the raised questions and concerns, and incorporated your constructive suggestions into the revised version. The major revisions are summarized:

*** Theoretical analysis in Section 2 ***

To better motivate the proposed 2TS model, as suggested by Reviewer 2, we provide a theoretical analysis of the benefits of the 2TS model in Section 2 and Appendix A. Although the setting for the analysis can not cover all aspects of real-world recommendation problems, it leads to clear insights on the 2TS design, and help us understand

(i) why the embeddings for active and inactive users should evolve in different time scales and two online learning (or adaptation) algorithms are needed, and

(ii) why a differential use of transductive and inductive embeddings for active and inactive users is expected to lead to better performance.

It will be very appreciated if reviewers could read Section 2 and reconsider this as another significant contribution of this paper.

***  Additional baselines ***

We include three additional baselines (GCMC-GAT, SAR4Rec, and Bert4Rec) in Table 1. Two of them (SAR4Rec and Bert4Rec) are suggested by Reviewer 1. Including these 3 baselines, we have compared our method to 12 baselines, covering four categories and including strong and advanced methods. **Apart from that, our experimental results are strong and convincing also from the following aspects:**

- We conduct experiments on the largest public datasets (Taobao and Movielens-25M). We believe this is particularly valuable for a recommendation paper, since (1) the current scale of industry problems are billion-level, and (2) it is clear that the performances of the same model on different scales dataset can be very different, as some recent advanced methods can barely scale to Taobao dataset without our modification to them. Therefore, our performances on Taobao and Movielens-25M should be viewed as very strong and convincing results.

- We have put a significant amount of effort into reproducing and even improving the performances of the baselines for a fair comparison. As indicated by the symbol * and # in Table 1, some released implementations have been modified to better adapt to the dataset and evaluation protocol, and some implementations which are not runnable on Taobao have also been modified to be more scalable by us.

- We promise that we will release the code upon acceptance, and we will provide runnable *.sh* files so that the performances of our model (as shown in Table 1) are reproducible.

*** Typos and figures ***

We have gone through the paper to correct the typos. We have also re-plotted/polished Figure 1, 2, and 3 to make them clearer.

---

### Decision · Program_Chairs · 2021-01-07
**Final Decision**

**Decision:**

Reject

**Comment:**

This paper received divergent scores (one strong negative and three positives). The positive reviews praise the clear intuition/motivation and strong empirical performance, while the negative review considers the proposed approach ad-hoc with limited novelty. I read the paper myself and found myself leaning more towards the negative score. In more details:

I think the paper proposed a cleverly-engineered solution to employ two separate RNNs to model two different subsets of the users (active v.s. inactive). To combat overfitting, the authors proposed some tricks: 1) use MF to learn a better initialization and 2) tie some of the parameters together. The ablation study shows that both are quite useful, and not too surprisingly when you have two powerful RNNs and work hard to make sure they don't overfit, they perform better than a single RNN. As I mentioned earlier, this whole approach is quite cleverly-engineered and executed. But it's not clear to me if this is something that the ICLR community can benefit from (maybe except a relatively small proportion). I believe this paper can find a bigger audience in venues like KDD whose deadline is coming up.

Furthermore, the authors presented some theoretical analysis to justify their intuition, which to me feels a bit forced. To start, a big assumption is that the optimal user embeddings are very concentrated, which is a rather strong assumption and I hardly believe it will hold in practice (what can even be considered as "concentrated" in high-dimensional space?). Following that, the theorem implies that the initial embedding for the inactive users should be the expected optimal embedding, but then later in the paper this point seemingly got completely ignored and instead the authors just proposed to learn a common initialization. Additionally, the theorem suggests that there is an optimal threshold, but later in the paper again this point got completely ignored. I know the theory makes assumptions and builds on simple cases. But my point is that you don't need theorem to show me that two RNNs can perform better than one when trained properly (which, I admit, is non-trivial and is the main contribution of the paper). There has been a lot of discussion around the ML community about the trend of making your paper look "mathy" and I don't think this is a good thing.

Minor comment: in MF, the objective wrt each embedding is convex, but the whole optimization is not jointly convex and it is not likely that you can get to the global optimum. It is relatively insensitive to initialization though (comparing with neural nets).